# Unravelling the age of fine roots of temperate and boreal forests

Emily F. Solly [1,4], Ivano Brunner[1], Heljä-Sisko Helmisaari[2], Claude Herzog[1], Jaana Leppälammi-Kujansuu[2], Ingo Schöning[3], Marion Schrumpf[3], Fritz H. Schweingruber[1], Susan E. Trumbore[3] & Frank Hagedorn[1]

Fine roots support the water and nutrient demands of plants and supply carbon to soils. Quantifying turnover times of fine roots is crucial for modeling soil organic matter dynamics and constraining carbon cycle–climate feedbacks. Here we challenge widely used isotope-based estimates suggesting the turnover of fine roots of trees to be as slow as a decade. By recording annual growth rings of roots from woody plant species, we show that mean chronological ages of fine roots vary from <1 to 12 years in temperate, boreal and sub-arctic forests. Radiocarbon dating reveals the same roots to be constructed from 10 ± 1 year (mean ± 1 SE) older carbon. This dramatic difference provides evidence for a time lag between plant carbon assimilation and production of fine roots, most likely due to internal carbon storage. The high root turnover documented here implies greater carbon inputs into soils than previously thought which has wide-ranging implications for quantifying ecosystem carbon allocation.

[1] Swiss Federal Institute for Forest, Snow and Landscape Research WSL, Zürcherstrasse 111, 8903 Birmensdorf, Switzerland. [2] Department of Forest Sciences, University of Helsinki, P.O. Box 27 ,00014 Helsinki, Finland. [3] Max Planck Institute for Biogeochemistry, Hans Knöll Strasse 10, 07745 Jena, Germany. [4]Present address: Department of Geography, University of Zurich, Winterthurerstrasse 190, 8057 Zurich, Switzerland. Correspondence and requests for materials should be addressed to E.F.S. (email: emily.solly@wsl.ch)

Fine roots are fundamental for plant life; they acquire water and nutrients from the soil and act as a major conduit of carbon (C) below ground. Although these functions are of global importance and a prerequisite for Earth-system models, we have a surprisingly limited knowledge on the lifetime of fine roots[1]. Inputs of C from roots into the soil are among the most uncertain components in forest ecosystems, with estimates of fine-root lifetimes ranging from months to decades[2–5]. These disparate pictures of fine-root dynamics have enormous implications for our understanding of plant functioning and of the belowground C cycle, for instance, estimating the persistence of organic compounds in living roots before they become available for decomposition and are partly converted to soil organic matter.

Studies which adopt the radiocarbon ($^{14}$C)-bomb peak to estimate the mean age of C, that is, the average time elapsed since C was originally fixed from the atmosphere, have often yielded decade-old C in fine roots of trees[2,6,7]. The mean $^{14}$C age of tree fine roots is thought to result from a mixture of newly produced roots containing C taken up from the atmosphere 1–2 years previously and of older living roots[2,8]. In comparison, root cameras (minirhizotrons), screen counting techniques or steady-state mass balance methods (dividing stock by production, e.g., using in-growth cores) indicate that fine roots grow and die within a few years[3,9]. The apparent disagreement between these two estimates of fine-root dynamics has been resolved by the current view that fine-root systems are not homogenous, but have a broad spectrum of lifetimes depending on function, branch order or anatomy. Each of the various techniques identifies different parts of this age spectrum[10,11]. Distal absorptive or

mycorrhizal roots are commonly observed to be of a more ephemeral nature as compared to proximal, transportive fine roots with larger diameters[12]. Despite this improved understanding, our most detailed estimates of C fluxes from fine roots to soils[8,13,14] still contain substantial uncertainties. These uncertainties relate to: the lifetime distribution of fine roots; root lifetimes as a function of environmental conditions and plant species; and the sources and ages of C transferred to roots as a result of growth and maintenance[14].

Here, we present a novel dendrological approach that allows us to unravel the chronological ages of individual fine roots with different diameter and functions. The approach can be applied to woody plant species that form discernible annual growth rings in climates with strong seasonality. In parallel, we took $^{14}$C measurements of the same roots to estimate the mean C ages of fine roots. The C age of a root is a measure for the time elapsed since the root C was fixed from the atmosphere[2]. This combined approach sheds new light on the hidden life of fine roots in forests and helps to reduce the current uncertainties.

For a range of forest sites spanning a large gradient in environmental conditions, including temperate and boreal forests as well as the sub-arctic treeline, fine roots (<2 mm) of different woody plant species were collected from intact soil cores. In these forest sites, woody species form recognizable growth rings[15,16]. Root samples of the dominant woody plant species were used for anatomical analysis of root cross-sections of three diameter classes: <0.5 mm, 0.5–1 mm and 1–2 mm. The chronological age of roots belonging to each diameter class was determined by counting the number of annual growth rings in the secondary xylem of several individual fine roots (Fig. 1). In

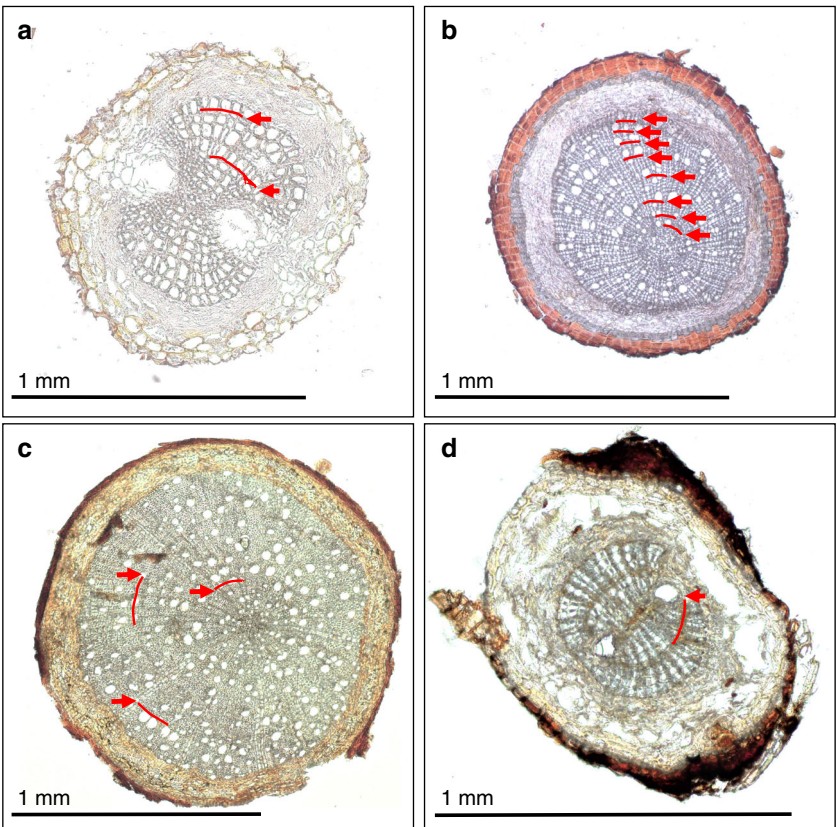

**Fig. 1** Examples of annual growth rings in the secondary xylem of fine roots. **a** *Pinus sylvestris* (pine), **b** *Betula nana* (birch), **c** *Fagus sylvatica* (beech) and **d** *Picea abies* (spruce). Red arrows denote transitions from latewood of the previous to earlywood of the following growing season. The black line represents the length of 1 mm. The ring boundaries of beech are more difficult to recognize as compared to the other studied species as they can be inconspicuous

| Forest type | Temperate | | | Boreal | Sub-arctic |
|---|---|---|---|---|---|
| Plant species | Pine | Beech | | Spruce | Birch |
| | Pfynwald | Schwäbische-Alb | Hainich-Dün | Flakaliden | Tchernaya |
| Location | SW Switzerland | SW Germany | Central Germany | NE Sweden | Northern Russia |
| Coordinates | 46°18′N, 7°37′E | 48°26′N, 9°23′E | 51°9′N, 10°28′E | 64°7′N, 19°27′E | 66°47′N, 66°51′E |
| Annual mean temperature | 9.2 °C | 6.0–7.0 °C | 5.5–8.0 °C | 1.2 °C | −3.9 to 4.7 °C |
| Annual mean precipitation | 657 mm | 700–1000 mm | 500–800 mm | 523 mm | 450–820 mm |
| Mean chronological age (no. of annual growth rings) | 1–2 (Data from this study) | 1.1–2.7 (Data from this study) | | 0.9–1.4 (Data from this study) | 4.4–11.8 (Data from this study) |
| Ingrowth cores and minirhizotrones (1/turnover rate) | 1.2–1.7[52] | 0.4–2.4[53,54] | | 0.7–2.0 1.0–2.1[19,55] | Data not available |
| Mean [14]C age (radiocarbon) | 6–14 (Data from this study) | 3–20 (Data from this study) | | 6–13 (Data from this study) | 15–23 (Data from this study) |

**Table 1 Site characteristics and comparison of fine root lifetimes using different methods**

Main characteristics of the four forest sites, and comparison of fine root (<2 mm diameter) lifetime ranges (in years) estimated by different methods in the four forest sites

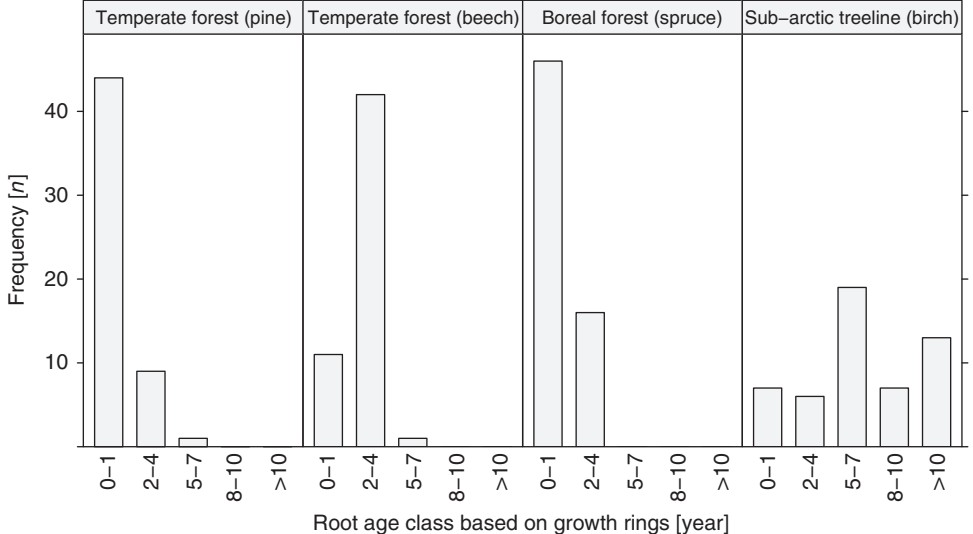

**Fig. 2** Age class distribution of fine roots estimated by counting the number of annual growth rings. The age class distribution of fine roots with a diameter size of <2 mm was estimated by counting the number of annual growth rings in the secondary xylem of fine-root samples of different climate zones with different dominant woody species

case no growth rings were present or a secondary xylem was absent, the roots were considered to be younger than one growing season. The applicability of the method was confirmed by the finding that yearly growth rings in roots of tree seedlings matched or did not exceed their known age, and that the maximum number of growth rings counted in newly produced roots grown in in-growth cores for 1 and 2 years was not more than one or two respectively. We estimated the mean chronological age of the overall fine root biomass <2 mm in each forest site using a weighted average. Hereby we consider the proportion of the root biomass that contributes to a given size class of fine roots (i.e., root biomass in a given size class/total weight of root biomass sample) and its respective average number of growth rings. The mean chronological age was then compared with the age of fine-root C estimated from [14]C measurements. The combination of these two approaches provides the first empirical evidence that across a wide range of forest ecosystems, the age of fine roots is significantly younger than that of the C used for the growth of fine-root structures.

This result has strong implications for estimating the flux input of C from fine roots into the soil.

## Results

**Mean chronological age of fine roots**. We observed that in temperate and boreal forests the mean chronological ages of fine roots ranged between 1 and 3 years (Table 1). At the sub-arctic treeline, living fine roots were significantly older (Fig. 2) ($P <$ 0.001) with mean chronological ages ranging between 4 and 12 years. The longer lifetime of birch fine roots at the sub-arctic treeline could be either an intrinsic property of this woody species or due to the cold climate with a short growing season. Under temperature-limited conditions, plants have to be more efficient with their C resources and invest more in root maintenance and longevity than in the formation of new roots[17–19]. The number of growth rings increased on average with diameter size (Fig. 3), but did not appear to be related to root branching order in root systems of tree seedlings (Supplementary Data 1).

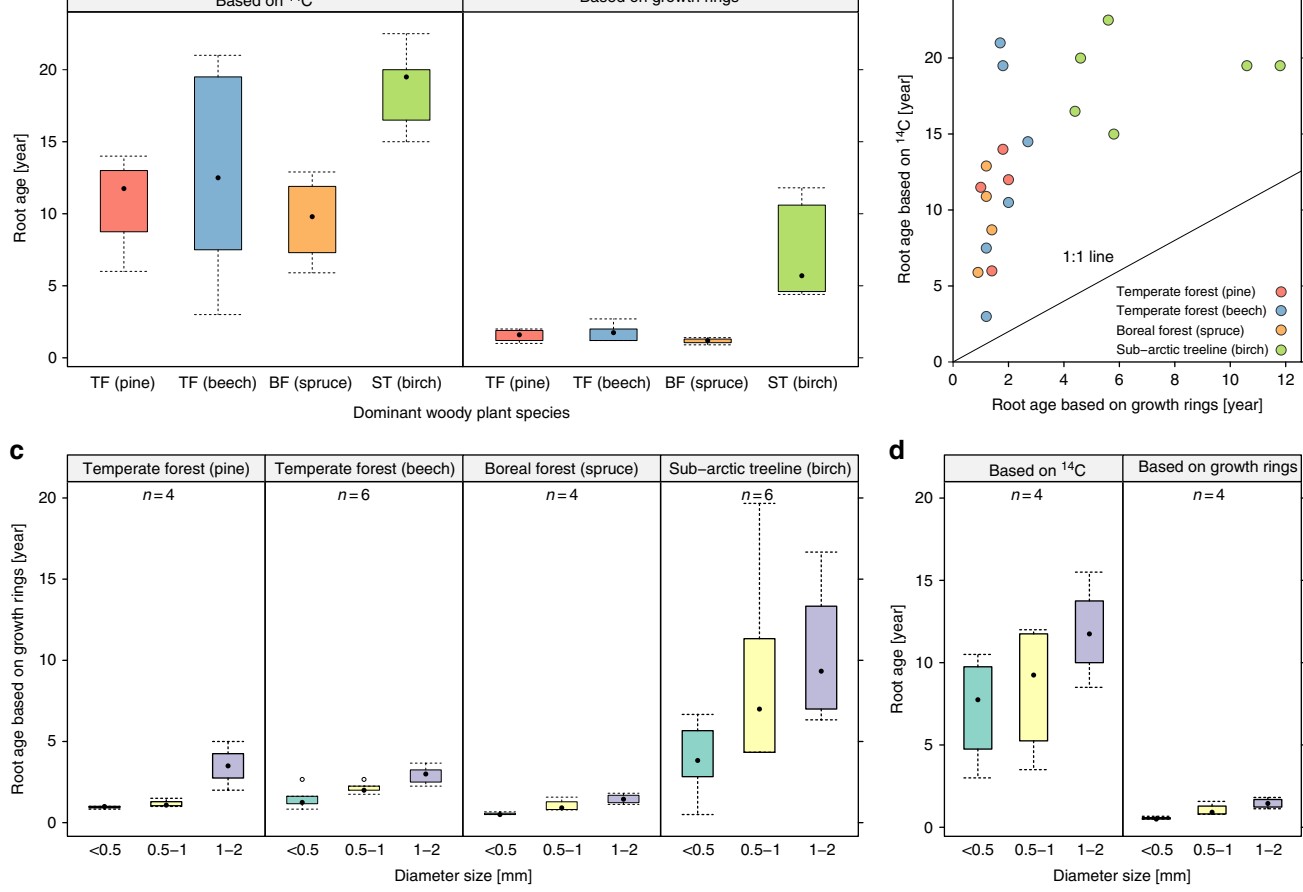

**Fig. 3** Comparison of root ages based on annual growth rings and based on radiocarbon. **a** Comparison between the age of fine roots (mixture of all fine roots with a diameter size <2 mm) estimated by counting the number of annual growth rings in the secondary xylem and the radiocarbon ($^{14}$C)-based age of the same fine-root samples in different climate zones with different dominant woody species, and **b** deviation from the 1:1 line. **c** Comparison between the age of fine roots with different diameter sizes (<0.5, 0.5–1, 1–2 mm) estimated by counting the number of annual growth rings among forest ecosystems. **d** In the boreal forest dominated by spruce, radiocarbon was also measured in roots with different diameter sizes (<0.5, 0.5–1, 1–2 mm) and compared to the age of fine roots estimated by counting the number of annual growth rings of the same fine root samples. In the box-plots, the dot represents the median, the box denotes the interquartile range, the whiskers show 1.5 times the interquartile range of the data and points more than 1.5 times the interquartile range are plotted individually[51]. TF temperate forest, BF boreal forest, ST sub-arctic treeline

**Comparison between chronological and $^{14}$C ages of fine roots.** On average, the chronological ages were $10 \pm 1$ (mean $\pm$ 1 SE) years younger than the mean $^{14}$C ages estimated for fine root C (Fig. 3), although we would expect the mean $^{14}$C ages to be younger than the age of roots, since most of the C post-dates the formation of the first growth ring. No significant difference was detected in the variance between the four forest types with different dominant woody species. The discrepancy between the lifetime of fine roots estimated by the two methods increased from $6.7 \pm 1.6$ over $7.4 \pm 2.0$ to $10.4 \pm 1.3$ years (mean $\pm$ 1 SE) for roots with diameter size <0.5, 0.5–1 and 1–2 mm, respectively, measured in the boreal forest dominated by spruce. Moreover, the $^{14}$C age of individual roots did not match their chronological age (Supplementary Data 1).

Previous studies suggested that an incomplete sampling of the youngest fine roots by destructive soil coring, as well as the difficulty of extracting the smallest fine roots from soil aggregates, might result in an overestimation of fine-root lifetimes via isotope measurements[11]. In our study, we probably did not capture all of the finest roots with faster than annual turnover with our sampling procedure. However, we used the same root biomass samples for both approaches, and the $^{14}$C age of individual roots

was still greater as their chronological age (Supplementary Data 1). These findings, thus, provide evidence that older C was used to build new roots.

## Discussion

The most plausible reason for the observed 2- to 19-year discrepancy in root ages between the methods ($P < 0.01$) is that there is a time lag between C acquisition through photosynthesis and the use of that same C for the formation of fine roots. Primary products of photosynthesis such as nonstructural carbohydrates may be stored for an extended time and only then be used for primary metabolism such as growth and development[20]. In most vascular plants, starch is known to play an important role in the day-to-day carbohydrate metabolism of the leaf, but further accumulates in non-photosynthetic tissues of plants, including seeds, stems and roots[21]. This stored starch is deposited in amyloplasts of heterotrophic cells serving as a medium- to long-term energy source to fuel growth processes. Decade-old C was unequivocally found to be used for the re-growth of stump sprouts[22], and trees were observed to allocate longer-lived storage C pools to the production of new fine roots after periods of

canopy defoliation and root mortality[23] as well as periods of drought[24]. For most perennial plant species and mature trees, however, it is still largely unknown whether allocation of new photosynthetic products to storage is an actively regulated process or merely a stochastic buffer that equilibrates disparities in demand and supply[25,26]. The recycling of older C from dying roots to support seasonal growth of new roots could similarly cause C to remain in the biomass of fine roots for a considerably longer time than in individual roots themselves[3,27].

Another explanation for the observed discrepancy could be that fine roots take up older C sources from soils and then use it to produce new root structures. Carbon and nutrient flow at the soil–root interface is bidirectional, with C being lost from roots as rhizodeposits, and simultaneously taken up from the soil, predominantly in the form of low molecular weight solutes and amino acids[28,29]. Roots may additionally take up inorganic C passively from outside the root when present in a dissolved form[30]. Whether this flux of C is of ecological significance for plant nutrition and metabolism is highly uncertain. We cannot rule out that older C compounds are taken up from soils or that year-to-decade old $CO_2$ in soil air space can diffuse in root tissues, and hence contribute to the divergence between $^{14}C$-based mean ages and chronological ages of fine roots. However, our data suggest that these mechanisms are unlikely to explain the full extent of the discrepancy. We found a systematic age difference in all the investigated sites, even in very acid soils (e.g., at the sub-arctic treeline in the Polar Urals) where very old $^{14}C$-free carbonates are not present (Supplementary Table 1). Moreover, in grasslands with a more active rhizosphere than in forests, fine roots were found to have mean $^{14}C$ ages of <1 to 5 years[6], indicating that uptake from old soil organic matter or as dissolved inorganic C is negligible. The observation that fine roots grown from young tree seedlings contain structural C equal or younger than the age of the seedlings themselves rather than older C (Supplementary Data 1) further corroborates that the uptake of aged C from soils is at most a minor contributor to the observed discrepancy between chronological and $^{14}C$-based fine-root ages.

Exchange of older C through root grafts and common mycorrhizal networks are alternative pathways which remain to be explored[31,32]. A re-translocation of organic substrates in the root–mycorrhiza system could potentially lead to the usage of older C for the construction of fine roots. Some ectomycorrhizal fungi have in fact been observed to take an active part in the decomposition of older organic matter to mine for nitrogen[33]. Nevertheless, the mobilization of nitrogen from soil organic matter by ectomycorrhizal fungi is regarded as a co-metabolic oxidation process[34], and the metabolic C demand of ectomycorrhizal fungi is likely not met by organic matter decomposition, but rather primarily supplied by host plants in exchange for nitrogen[34]. Consequently, the large incongruity between chronological and $^{14}C$-based mean ages observed in our study is unlikely to be related to the transfer of C between mycorrhiza and roots. Radiocarbon measurements of mycorrhizal sporocarps suggested that the $^{14}C$ age of fungal symbionts resembles current year needles, but that C sources other than atmospheric $CO_2$, such as stored carbohydrates, may also contribute small amounts of C[35]. Comparing the $^{14}C$ age of fine roots and of their non-structural carbohydrates with the $^{14}C$ age of mycorrhizal fungal sporocarps over the course of 1 year, or studying the cycling of C in the root–mycorrhiza system with isotopic labeling techniques, might help to refine the mechanisms of plant C allocation.

With our approach, we resolve the discrepancies between fine-root lifetimes estimated from $^{14}C$ measurements and other methods used to study fine-root dynamics. For the first time, we

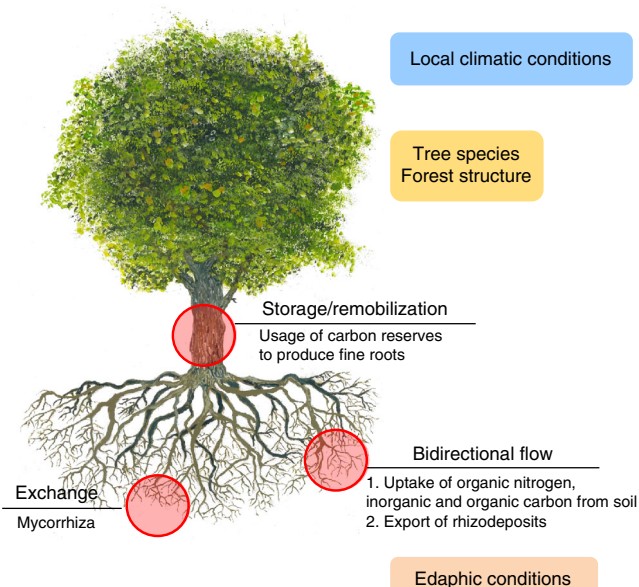

**Fig. 4** Summary of possible drivers explaining the difference between the chronological age and the mean radiocarbon age of living fine roots. Drivers can be biotic (e.g., forest structure and intrinsic traits of plant species) and abiotic (e.g., climatic conditions and edaphic properties). These factors can alter the amount of old vs new carbon reserves used by trees and other perennial plants for growth and maintenance of fine roots, consequently influencing their $^{14}C$ age. The exchange of organic substrates in root–mycorrhiza systems and the uptake of resources from the soil may alter the lifetime of fine roots and influence their mean $^{14}C$ age. Tree graphic used with the permission of the creator; Hanspeter Läser. The copyright remains with the creator and is not affected by this article's CC BY licence

successfully controlled space and time by measuring the chronological age of individual fine roots by counting the number of annual growth rings in root tissues of woody plants in parallel to their mean $^{14}C$ ages. Our results demonstrate that, across a wide range of forest ecosystems from temperate forests to the sub-arctic treeline, the age of fine roots is substantially younger than that of the C used for their growth. This finding has profound implications for the understanding of C dynamics in the plant and soil system. Firstly, trees are apparently using years to decade older internally stored C to form new roots, implying that the upper-end of fine-root lifetimes—commonly derived from C isotope measurements in fine roots—is several years shorter than previously estimated. Secondly, without considering the time between C is fixed by plants and is used for root growth, radiocarbon-based studies quantifying fine-root lifetimes could largely underestimate the input of C from fine roots into the soil.

All in all, the finding of older C in chronologically younger roots challenges the conventional paradigm that different measurement techniques gather information about different ends of the root-lifetime continuum. How long C taken up via photosynthesis resides in the plant before it is used for fine-root formation likely depends on intrinsic traits of plant species and/or their adaption to local environmental conditions (Fig. 4). Understanding the mechanisms and regulating factors of this C use of woody species for fine-root growth will provide novel insights into the C allocation of plants and will advance our understanding of above- and belowground C cycling.

## Methods

**Forest sites.** In this study, fine-root ages were assessed in four types of forests in three climatic zones. The forests included: a temperate coniferous forest in Switzerland (Pfynwald) dominated by Scots pine (*Pinus sylvestris* L., referred to as pine), temperate broadleaved forests in Germany (Hainich-Dün and Schwäbische-Alb) dominated by European beech (*Fagus sylvatica* L., referred to as beech), a boreal coniferous forest in Sweden (Flakaliden) dominated by Norway spruce (*Picea abies* (L.) Karst, referred to as spruce) and a sub-arctic treeline forest in the Polar Urals (Tchernaya) mountains in Russia, where the root biomass was dominated by roots belonging to dwarf birch (*Betula nana* L., referred to as birch). The four types of forest span a gradient in mean annual temperature from −3.9 to 4.9 °C to 9.2 °C. Details on site and soil properties are given in Table 1 and Supplementary Table 1.

**Sample collection.** Fine roots <2 mm were collected down to the upper 10 cm of the mineral soil, where the vast majority of the roots were found for some of the selected forest sites. This depth comprised 97% of fine roots in the birch forest at the sub-arctic treeline[36], as well as in the pine forest in Switzerland[24], and 86% of fine roots in the boreal forest dominated by spruce[37]. For comparability of measurements between the four sites we selected a consistent sampling depth of 0–10 cm in the mineral soil. In the temperate coniferous forest in Switzerland, fine roots were sampled in April 2003 in four plots (25 × 40 m). Two separate soil monoliths were taken at a distance of 0.5–1 m from three trees within each plot, by use of a cylindrical corer (4.5 cm in diameter)[38]. In the temperate broadleaved forests in Germany, fine roots were sampled in May 2011 from five forest sites in the Schwäbische-Alb and one forest site in the Hainich-Dün. In each site, 14 mineral soil monoliths were collected along two 40 m transects, with a cylindrical corer (5 cm in diameter), and mixed to make a composite sample[6]. In the boreal coniferous forest in Sweden, fine roots were collected from three soil monoliths per plot in June 2009. Soil monoliths were randomly sampled from two irrigated and two warmed-irrigated plots at the long-term experimental site of Flakaliden, by use of a cylindrical soil corer (3.9 cm in diameter)[37,39]. In the stone-rich sub-arctic treeline in the Polar Ural mountains, fine roots were collected in 2004 from three different elevations (200, 237 and 309 m above sea level) by excavating one quantitative soil pit with an area of approximately 20 × 20 cm. Soil pits were randomly excavated within a 1 × 1 m plot under a tree (>2 m height) at each elevation. The total soil material was sampled in each plot and its volume was determined by measuring the dimension of the soil pit with a ruler and by filling the hole with a known volume of sand. Fine roots were then carefully picked out from the collected soil as in ref. [36].

**Fine root biomass.** Fine roots <2 mm were manually sorted from the soil. The mineral-soil particles attached to the roots were carefully removed with deionized water. Living fine roots belonging to the dominant woody species in each forest site were sorted from dead roots according to their color, cohesion of cortex and viscosity. For anatomical analysis, the root samples were further divided into three diameter classes: diameter <0.5 mm, diameter 0.5–1 mm and diameter 1–2 mm. The samples were dried to constant weight and their biomass was weighed. The dry root biomass of the three diameter sizes was also assessed for each sample. As fine-root samples were not frozen at −80 °C or shock heated with microwaves immediately after the sampling, it was not possible to measure nonstructural carbohydrates for this study[40,41].

**Root anatomy.** Pine, beech, spruce and birch form discernible boundaries between two neighboring growth increments[15,16]. In regions where climate exhibits a strong annual periodicity, these boundaries are used in dendroecological studies to determine tree ages and to reconstruct climatic conditions (annual growth rings). Here in our study, we counted the growth rings in fine roots for all samples by randomly choosing at least 3–4 individual root segments for each diameter size class. Some samples did not have sufficient roots for analysis in some of the diameter classes. Fine-root thin sections (approximately 15–20 μm) were prepared using a laboratory microtome[42], covered with glycerol and a cover glass, and photographed with a BX41 system microscope (Olympus, Tokyo, Japan). The number of growth rings was counted in the secondary xylem in all the thin sections of the individual fine-root segments. Since in roots the latewood often consists solely of a single row of cells, both complete and incomplete growth rings can be present and tapering radial growth can occur, only the complete growth rings were taken into account during the measurements. In case no growth rings were present, or a secondary xylem was absent, the roots were considered to be younger than one growing season (that is, 0.5 years). The robustness of the method was tested by counting the maximum yearly growth rings in roots of tree seedlings (*Pinus sylvestris*, *Fagus sylvatica* and *Picea abies*) of known age grown from seeds in the garden of the Swiss Federal Institute for Forest, Snow and Landscape Research WSL (see example in Supplementary Fig. 1), and by counting the maximum number of growth rings of *Pinus sylvestris* fine roots grown in ingrowth cores which were kept in the soil of the temperate coniferous forest in Switzerland for 1 and 2 years, as described in ref. [38]. In both analyses, the number of root rings reflected the known ages (Supplementary Data 1), which is in agreement with the finding that growth rings in the roots of temperate perennial forbs are robust

annual markers[43]. The chronological age was seldom as much as 2 years in doubt and probably more frequently overestimated than underestimated. Note that this approach cannot be applied to woody fine roots of tropical forests, because they do not contain any annual growth rings as climate fluctuations have no strong annual rhythm[44].

**Radiocarbon analysis.** Radiocarbon ([14]C) values of ground fine-root biomass samples from the different forest types were analyzed as follows. The pine roots from temperate forest were analyzed at the laboratory of Ion Beam Physics (ETH Zürich, Switzerland) using the accelerator mass spectrometry (AMS) system of MICADAS[45]. Fine-root samples (bulked for roots <2 mm) were prepared as described in ref. [24]. The beech roots from temperate forest were analyzed at the [14]C analysis facility in Jena, Germany, using the AMS system 3MV Tandetron accelerator[46]. The preparation of fine-root samples (bulked for roots <2 mm) is described in ref. [6]. Spruce roots from boreal forest were analyzed at the NERC Radiocarbon Facility and SUERC AMS Laboratory, East Kilbride, UK[47]. The preparation of fine-root samples is described in refs.[7,39]. [14]C was measured in root samples belonging to three diameter classes (<0.5 mm, 0.5–1 mm and 1–2 mm)[39]. For comparison with the other forest sites, weighed mean [14]C ages of fine-root samples (combined for roots <2 mm) were calculated for each experimental plot by taking the individual values for each diameter class and considering the proportion of the root biomass contributing to every size class (Supplementary Fig. 2). Birch roots from sub-arctic treeline were analyzed at the [14]C analysis facility in Jena, Germany, using the AMS system 3MV Tandetron accelerator[46]. The preparation of fine-root samples (bulked at <2 mm) was done as described in ref. [6].

Data are reported as fraction modern: the [14]C/[12]C ratio in the sample, corrected for mass-dependent isotope fractionation to a common $\delta^{13}C$ value of −25‰, divided by that of an absolute [14]C standard (95% of the [14]C/[12]C ratio of the oxalic acid I standard with $\delta^{13}C$ of −19 ‰). We used [14]C to estimate root C age by use of the calibration program CALIBomb http://calib.qub.ac.uk/CALIBomb/ (accessed December 2017), with the IntCal13 pre-Bomb Calibration data set and the data and the Levin post-Bomb Calibration data set[48]. For a thorough comparison we also used the calibration program OxCal 4.2[49], using the Northern Hemisphere Zone 1 data set[50]. The mean C age was calculated for the [14]C values of the fine-root biomass samples, considering the distribution of the oldest and the youngest calibrated dates from both programs. Calculations are presented in Supplementary Data 1.

Individual roots with contrasting number of growth rings were also measured for [14]C at the [14]C analysis facility in Jena, Germany (Supplementary Data 1). For four individual *Pinus sylvestris* roots with a diameter of 1–2 mm, the inner woody part ("core") was further separated from the bark of a segment of the roots.

**Statistics.** The statistical difference between the mean ages of fine roots based on [14]C and those determined by counting the number of annual growth rings was tested with the Wilcoxon signed rank test. One-way analysis of variance (ANOVA) accompanied by Tukey's multiple comparisons of means at 95% family-wise confidence level was used to detect significant differences of mean fine-root ages determined by both methods, as well as their arithmetical difference, among forest types with different dominant woody plant species. To detect statistical differences in the mean ages of fine roots determined by counting the number of annual growth rings, for roots with different diameter sizes and across the different forest types, one-way and two-way ANOVA accompanied by Tukey's multiple comparisons of means at 95% family-wise confidence level was adopted respectively. Prior to analysis, the mean ages of fine roots determined by counting the number of annual growth rings were log10 transformed to meet assumptions of normality. For all statistical tests, a significance level of 0.05 was selected. All analyses were conducted with the R version 3.2.3[51], see Supplementary Software 1.

**Data availability.** The data that support the findings of this study are provided in Supplementary Data 1.

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

## Acknowledgements

This research was supported by the Swiss Federal Research Institute WSL. The Pfynwald forest is part of the Swiss Long-term Forest Ecosystem Research program LWF (www.lwf.ch), which is part of the UNECE Co-operative Programme on Assessment and Monitoring of Air Pollution Effects on Forests ICP Forests (www.icp-forests.net). Main funding sources included the Swiss National Science Foundation (grant SNF 31003A_149507), the DFG Priority Program 1374 Infrastructure- Biodiversity-Exploratories (SCHR 1181/2-1), the INTAS (grants 01-0052 and 04-83-3788) and COST Action ES1203 SENFOR (SBFI No. C14.0037). We gratefully thank Axel Steinhof for the radiocarbon analysis at the AMS facility in Jena, Germany, and the Academy of Finland (grant 260708) for funding the collection of the Flakaliden samples and their radiocarbon analysis at the NERC Radiocarbon Facility, Scotland. We thank Sonja G. Keel for the helpful discussions, and Sonia Meller for her contribution of some fine-root samples.

## Author contributions

E.F.S., I.B., F.H.S. and F.H. designed the study. I.B. and C.H. provided pine root samples and performed their radiocarbon analysis. H.-S.H. and J.L.-K. provided spruce root samples and performed their radiocarbon analysis. E.F.S., I.S., M.S. and S.E.T. provided beech root samples and performed their radiocarbon analysis. F.H. provided birch root samples. S.E.T.

and M.S. performed the radiocarbon analysis of birch root samples and of single individual roots. E.F.S. and F.H.S. performed the root anatomical analysis. E.F.S. analyzed the data. E.F.S. wrote the manuscript and all authors contributed to reviewing the manuscript.

## Additional information

**Competing interests:** The authors declare no competing interests.

