## [Peer Review File · Nature Communications]

Reviewers' comments:

Reviewer #1 (Remarks to the Author):

Great manuscript! Clearly written, novel and of clear interest for anyone studying root systems. The rationale of comparing the root age of roots of known ages by counting the growth rings, and then to analyse the isotopic age is a straightforward and convincing lead. I have only two comments focussing on two passages of the manuscript that puzzled me somewhat (neither one is critical relative to the set-up or conclusions of the paper):

1. In the discussion, you write: "Mycorrhizal fungal sporocarps have also been found to resemble current year needles or atmospheric CO₂. Consequently, the large incongruity observed in our study is unlikely to be related to the re-translocation of C between mycorrhiza and roots." What are the implications of this? Is fresh carbon transferred directly after photosynthesis to the mycorrhizal partners, and is only old(er) carbon used for fine root growth? Are these fluxes then separated within the roots? Or do the fluxes of carbon from photosynthesis and those for root growth occur at different moments in the season?

2. You state some ten lines lower: 'Secondly, our results give evidence that pre-aged C is transferred into the soil and thus, inputs of root-derived C into soils are substantially greater than assumed by isotopic studies.' What do you mean with pre-aged C and why does your study give evidence that isotopic studies underestimate carbon fluxes from roots into the soil? It could simply be that the carbon stays longer in the root system before it passes into the soil.

Reviewer #2 (Remarks to the Author):

The MS titled "Unravelling the age of fine roots of temperate and boreal forests" suggests that relatively older carbon from storage sources is used to build new fine roots and therefore previous research using C¹⁴ to estimate the age of fine roots is inaccurate. Fine roots support the water and nutrient demands of plants and supply carbon to soils. Quantifying turnovers of fine roots is crucial for our understanding of global carbon cycle.

Overall, I think the MS is well written and serves as an important contribution to the scientific community in general.

The authors used annual growth rings of fine roots from four different woody plant species from different ecosystems temperate, boreal and subarctic forests. The method used is a novel dendrological approach that allows to unravel the "chronological ages" of individual fine roots with different diameter and functions. Chronological ages of the fine roots studied vary from <1 to 12 years in the different ecosystems. Radiocarbon dating of the same root samples showed that roots were older than what was seen from the anatomy analysis of the rings within the fine roots. The authors conclude that the difference provides evidence for a time lag between plant carbon assimilation and production of fine roots, most likely due to internal carbon storage and therefore older C was used to build new roots and there is a time lag between C acquisition through photosynthesis and the use of that same C for the formation of fine roots.

The authors did not measure primary products of photosynthesis such as nonstructural carbohydrates (NSC) in the species examined. Carbon dating of the NSC could have helped in reaching a better understanding of the process. If such an approach was not optional or helpful the authors' in my opinion, should discuss it.

However, I agree with the authors statement that the large incongruity observed in their study is unlikely to be related to the re-translocation of C between mycorrhiza and roots and that it is probably a result of delayed carbon allocation. This statement was well discussed in the MS.

The authors did not explain why they sampled roots from depth of up to 10cm and not, for

instance, from 30cm which is a good representative for the majority of roots in the studies ecosystems.

In addition to their main finding the authors also compare between the age of fine roots from the different ecosystem. This by itself is important, however their explanations for these differences such as the longer lifetime of birch fine roots at the sub-arctic that under temperature-limited conditions, plants have to be more efficient with their C resources and invest more in root maintenance and longevity than in the formation of new roots must be backed up with references.

As for the figure presentation:

- In Figure 3 the figure within the figure is not clear and is not related to in the figure legend.
- In figure 2A and B it is not clear whether the results were obtained from a mixture of root or from a specific size.
- In Figure 2D it is not clear why only results from spruce are shown. Maybe it's better to replace figures C and D in one figure that related each species.
- Supplementary table 2 is key for the paper and authors may consider adding it to the main text

NCOMMS-18-06673 Unravelling the age of fine roots of temperate and boreal forests
by Emily F. Solly, Ivano Brunner, Heljä-Sisko Helmisaari, Claude Herzog, Jaana Leppälammil-Kujansuu, Ingo Schöning, Marion Schrumpp, Fritz H. Schweingruber, Susan E. Trumbore, Frank Hagedorn

Response to Reviewers:

We thank both reviewers for their time and appreciate their favourable responses as well as helpful comments and suggestions to improve the manuscript.

Reviewer #1 (Remarks to the Author):

Great manuscript! Clearly written, novel and of clear interest for anyone studying root systems. The rationale of comparing the root age of roots of known ages by counting the growth rings, and then to analyse the isotopic age is a straightforward and convincing lead. I have only two comments focussing on two passages of the manuscript that puzzled me somewhat (neither one is critical relative to the set-up or conclusions of the paper):

1. In the discussion, you write: “Mycorrhizal fungal sporocarps have also been found to resemble current year needles or atmospheric CO₂. Consequently, the large incongruity observed in our study is unlikely to be related to the re-translocation of C between mycorrhiza and roots.” What are the implications of this? Is fresh carbon transferred directly after photosynthesis to the mycorrhizal partners, and is only old(er) carbon used for fine root growth? Are these fluxes then separated within the roots? Or do the fluxes of carbon from photosynthesis and those for root growth occur at different moments in the season?

Thank you for this interesting comment to consider and address the implication of this statement. Our study does not provide evidence for a potential splitting of carbon (C) sources between roots and mycorrhizal partners as we have not measured radiocarbon (¹⁴C) in associated fungi and we only referred to other studies indicating that C from mycorrhizal fungi is predominantly young¹ and thus, very unlikely contributes to the relative ‘old’ ¹⁴C age of roots. However, we know that for many tree species there is a first flush of root growth already before foliage development^{2,3}. This would require the usage of storage compounds for fine-root growth, while more recent photosynthetic products could be used later. Thus, we primarily relate the potentially differing ¹⁴C age between fine roots and mycorrhizal symbionts to seasonally varying sources of C. However, in order to proof this mechanism, it would be necessary to repeatedly measure ¹⁴C in fine-roots and their storage compounds as well as to measure the ¹⁴C age of mycorrhizal fungal sporocarps during a whole year. An alternative method would be to study the cycling of C in the root – mycorrhizal system with isotopic-labeling techniques. In the revised manuscript, we have added the following section to explain the mechanism but also ways of how to tackle the transfer of C between fine roots and their mycorrhizal partners. “Exchange of older C through root grafts and common mycorrhizal networks are alternative pathways which remain to be explored^{4,5}. A re-translocation of organic substrates in the root-mycorrhiza system could potentially lead to the usage of older C for the construction of fine roots. Some ectomycorrhizal fungi have in fact been observed to take an active part in the decomposition of older organic matter, to mine for nitrogen⁶. Nevertheless, the mobilization of nitrogen from soil organic matter by ectomycorrhizal fungi is regarded as a co-metabolic oxidation process⁷, and the metabolic C demand of ectomycorrhizal fungi is likely not met by organic matter decomposition, but rather primarily supplied by host plants in exchange for nitrogen⁷. Consequently, the large incongruity between chronological and ¹⁴C-based mean ages observed in our study is unlikely to be related to the transfer of C between mycorrhiza and roots. Radiocarbon measurements of mycorrhizal sporocarps suggested that the C age of fungal symbionts resembles current year needles, but that C sources other than atmospheric ¹⁴CO₂, such as stored carbohydrates, may also contribute small amounts of C¹. Comparing the ¹⁴C age of fine roots and of their nonstructural carbohydrates with the ¹⁴C age of mycorrhizal fungal sporocarps over

the course of one year, or studying the cycling of C in the root - mycorrhiza system with isotopic labeling techniques, might help to refine the mechanisms of plant-C allocation”.

2. You state some ten lines lower: ‘Secondly, our results give evidence that pre-aged C is transferred into the soil and thus, inputs of root-derived C into soils are substantially greater than assumed by isotopic studies.’ What do you mean with pre-aged C and why does your study give evidence that isotopic studies underestimate carbon fluxes from roots into the soil? It could simply be that the carbon stays longer in the root system before it passes into the soil.

We acknowledge that it is important to clarify and modify this sentence in the manuscript, and we have eliminated the term ‘pre-aged’. We observed that the chronological age of fine roots in temperate, boreal and sub-arctic forests is several years younger than the C used for fine-root growth (based on the ¹⁴C measurements). This finding suggests an overestimation of fine-root lifetimes based on ¹⁴C data in forest ecosystems, most likely due to the use of older C from storage sources of plants to grow new fine roots. Therefore, without considering the time between C fixation by plants and the time it is used to grow root structures, radiocarbon-based studies quantifying fine-root lifetimes could significantly underestimate the input of C from fine roots into the soil. We have modified this sentence in the manuscript: “Secondly, our findings indicate that without considering the time between C fixation by plants and the time it is used for root growth, radiocarbon-based studies quantifying fine-root lifetimes could largely underestimate the input of C from fine roots into the soil”.

Reviewer #2 (Remarks to the Author):

The MS titled “Unravelling the age of fine roots of temperate and boreal forests” suggests that relatively older carbon from storage sources is used to build new fine roots and therefore previous research using C14 to estimate the age of fine roots is inaccurate. Fine roots support the water and nutrient demands of plants and supply carbon to soils. Quantifying turnovers of fine roots is crucial for our understanding of global carbon cycle.

Overall, I think the MS is well written and serves as an important contribution to the scientific community in general.

We are pleased about the favourable response and we thank the reviewer for the comments provided below, which improve the explanation of our results.

The authors used annual growth rings of fine roots from four different woody plant species from different ecosystems temperate, boreal and subarctic forests. The method used is a novel dendrological approach that allows to unravel the “chronological ages” of individual fine roots with different diameter and functions. Chronological ages of the fine roots studied vary from <1 to 12 years in the different ecosystems. Radiocarbon dating of the same root samples showed that roots were older than what was seen from the anatomy analysis of the rings within the fine roots. The authors conclude that the difference provides evidence for a time lag between plant carbon assimilation and production of fine roots, most likely due to internal carbon storage and therefore older C was used to build new roots and there is a time lag between C acquisition through photosynthesis and the use of that same C for the formation of fine roots.

The authors did not measure primary products of photosynthesis such as nonstructural carbohydrates (NSC) in the species examined. Carbon dating of the NSC could have helped in reaching a better understanding of the process. If such an approach was not optional or helpful the authors’ in my opinion, should discuss it.

It is true that we did not investigate nonstructural carbohydrates (NSC) for this study. The main reason for this is that the study relied on existing root samples from different forest ecosystems. All of our samples have been dried to constant weight, while for NSCs plant samples need to be frozen at -80°C or

to be shock heated with microwaves within a very short time after the sampling to prevent a further transformation of NSC^{8,9}. In any case, we agree with the reviewer that it would be an important step forward to provide ¹⁴C dating of NSCs in root tissues of plant species – such analyses are extremely rare in the literature. To date we are aware only of one study that measured starch and sugars in fine roots of two temperate trees⁸. We have added the following sentence in the text: “As fine-root samples were not frozen at -80°C nor shock heated with microwaves immediately after the sampling, it was not possible to measure nonstructural carbohydrates for this study^{8,9}”. Moreover, we have now added a sentence about the importance of measuring nonstructural carbohydrates in fine roots repeatedly throughout the year in the discussion section. “Comparing the ¹⁴C age of fine roots and of their nonstructural carbohydrates with the ¹⁴C age of mycorrhizal fungal sporocarps over the course of one year, or studying the cycling of C in the root - mycorrhiza system with isotopic labeling techniques, might help to refine the mechanisms of plant-C allocation”.

However, I agree with the authors statement that the large incongruity observed in their study is unlikely to be related to the re-translocation of C between mycorrhiza and roots and that it is probably a result of delayed carbon allocation. This statement was well discussed in the MS.

The authors did not explain why they sampled roots from depth of up to 10cm and not, for instance, from 30cm which is a good representative for the majority of roots in the studies ecosystems.

We acknowledge the need to specify why the roots were sampled from a depth of up to 10 cm. “Fine roots < 2 mm were collected down to the upper 10 cm of the mineral soil, where the vast majority of the roots were found for some of the selected forest sites. This depth comprised 97% of fine roots in the birch forest at the sub-arctic treeline¹⁰, as well as in the pine forest in Switzerland¹¹, and 86% of fine roots in the boreal forest dominated by spruce¹². For comparability of measurements between the four sites we selected a consistent sampling depth of 0-10 cm in the mineral soil”. We have added this information in the Methods section.

In addition to their main finding the authors also compare between the age of fine roots from the different ecosystem. This by itself is important, however their explanations for these differences such as the longer lifetime of birch fine roots at the sub-arctic that under temperature-limited conditions, plants have to be more efficient with their C resources and invest more in root maintenance and longevity than in the formation of new roots must be backed up with references.

We thank the reviewer for this comment. It is important to back up our explanation of the longer lifetime of birch fine roots with references. We now cite the paper by Gill and Jackson¹³ who observed a positive relationship between mean annual temperature and root turnover, as well as the paper by Finér, et al.¹⁴ whose results indicated that fine-root turnover decreases with latitude (smaller root turnover = greater root lifetime). In addition, Leppälammil-Kujansuu, et al.¹⁵ reported that spruce fine roots at a north boreal site had significantly longer lifetimes than spruce roots at a south boreal site. It has been suggested that these patterns could be explained by lower nutrient mineralization rates in cold regions and the necessity of plants to optimize their root maintenance for the uptake of nutrients, e.g. by reducing C expenses in fine root production and to prevent nutrient losses.

As for the figure presentation:

- *In Figure 3 the figure within the figure is not clear and is not related to in the figure legend.*

We agree that the figure within the figure did not add additional information to figure 3 and that was not well related to the description in the figure caption. Hence, we have now removed this part of the figure.

- *In figure 2A and B it is not clear whether the results were obtained from a mixture of root or from a specific size.*

In figure 2A and B the results were obtained from the combined fine roots with a diameter size < 2mm picked from the soil cores. We have specified this in the figure caption.

• In Figure 2D it is not clear why only results from spruce are shown. Maybe it's better to replace figures C and D in one figure that related each species.

While for fine roots of pine, beech and birch ^{14}C was measured in the root samples containing all fine roots with diameter < 2 mm, for fine roots of spruce ^{14}C was additionally measured in root samples belonging to three diameter classes (<0.5 mm, 0.5-1 mm and 1-2 mm). This approach enabled us to show that a large discrepancy exists between the age of fine roots estimated by counting the annual growth rings and by the ^{14}C measurements for fine roots belonging to different diameter sizes. We would hence prefer to keep Figure 2C and 2D as separate figures, but have added the following sentence in the caption for clarification. "In the boreal forest dominated by spruce, radiocarbon was also measured in roots with different diameter sizes (<0.5, 0.5-1, 1-2 mm) and compared to the age of fine roots estimated by counting the number of annual growth rings of the same fine root samples".

• Supplementary table 2 is key for the paper and authors may consider adding it to the main text

Thank you for this comment which will improve the manuscript. We have now moved Supplementary table 2 to the main text.

References:

- 1 Hobbie, E. A., Weber, N. S., Trappe, J. M. & Van Klinken, G. J. Using radiocarbon to determine the mycorrhizal status of fungi. *New Phytologist* **156**, 129-136 (2002).
- 2 Atkinson, D. The distribution and effectiveness of the roots of tree crops. *Horticultural Reviews, Volume 2*, 424-490 (2011).
- 3 Zamski, E. & Schaffer, A. A. *Photoassimilate distribution in plants and crops: source-sink relationships.*, 905 pp (Marcel Dekker Inc., 1996).
- 4 Salomón, R. L., Tarroux, E. & DesRochers, A. Natural root grafting in *Picea mariana* to cope with spruce budworm outbreaks. *Canadian Journal of Forest Research* **46**, 1059-1066 (2016).
- 5 Philip, L., Simard, S. & Jones, M. Pathways for below-ground carbon transfer between paper birch and Douglas-fir seedlings. *Plant Ecology & Diversity* **3**, 221-233 (2010).
- 6 Talbot, J. M. *et al.* Independent roles of ectomycorrhizal and saprotrophic communities in soil organic matter decomposition. *Soil Biology and Biochemistry* **57**, 282-291 (2013).
- 7 Lindahl, B. D. & Tunlid, A. Ectomycorrhizal fungi—potential organic matter decomposers, yet not saprotrophs. *New Phytologist* **205**, 1443-1447 (2015).
- 8 Richardson, A. D. *et al.* Distribution and mixing of old and new nonstructural carbon in two temperate trees. *New Phytologist* **206**, 590-597 (2015).
- 9 Bachofen, C., Moser, B., Hoch, G., Ghazoul, J. & Wohlgemuth, T. No carbon “bet hedging” in pine seedlings under prolonged summer drought and elevated CO_2 . *Journal of Ecology* **106**, 31-46 (2018).
- 10 Solly, E. F. *et al.* Treeline advances and associated shifts in the ground vegetation alter fine root dynamics and mycelia production in the South and Polar Urals. *Oecologia* **183**, 571-586 (2017).
- 11 Herzog, C., Steffen, J., Pannatier, E. G., Hajdas, I. & Brunner, I. Nine years of irrigation cause vegetation and fine root shifts in a water-limited pine forest. *PloS one* **9**, e96321 (2014).
- 12 Leppälammil-Kujansuu, J. *et al.* Effects of long-term temperature and nutrient manipulation on Norway spruce fine roots and mycelia production. *Plant and Soil* **366**, 287-303 (2013).
- 13 Gill, R. A. & Jackson, R. B. Global patterns of root turnover for terrestrial ecosystems. *The New Phytologist* **147**, 13-31 (2000).
- 14 Finér, L., Ohashi, M., Noguchi, K. & Hirano, Y. Fine root production and turnover in forest ecosystems in relation to stand and environmental characteristics. *Forest Ecology and Management* **262**, 2008-2023 (2011).
- 15 Leppälammil-Kujansuu, J. *et al.* Fine root longevity and carbon input into soil from below-and aboveground litter in climatically contrasting forests. *Forest Ecology and Management* **326**, 79-90 (2014).

REVIEWERS' COMMENTS:

Reviewer #1 (Remarks to the Author):

I am fully satisfied with the answers to the questions raised and the modifications applied to the manuscript, both to the comments I had and those of the other reviewer. A really fine manuscript! I am in favour of full acceptance.

Reviewer #2 (Remarks to the Author):

It seems the authors addressed the comments, nice work.

RESPONSE TO REVIEWERS' COMMENTS:

Reviewer #1 (Remarks to the Author):

I am fully satisfied with the answers to the questions raised and the modifications applied to the manuscript, both to the comments I had and those of the other reviewer. A really fine manuscript! I am in favour of full acceptance.

Reviewer #2 (Remarks to the Author):

It seems the authors addressed the comments, nice work.

We thank the reviewers for their positive feedback!